# Dynamic Change of Volatile Fatty Acid Derivatives (VFADs) and Their Related Genes Analysis during Innovative Black Tea Processing

**DOI:** 10.3390/foods13193108

**Published:** 2024-09-28

**Authors:** Zi-Wei Zhou, Qing-Yang Wu, Yang Wu, Ting-Ting Deng, Yu-Qing Li, Li-Qun Tang, Ji-Hang He, Yun Sun

**Affiliations:** 1College of Bioscience and Engineering, Ningde Normal University, Ningde 352000, China; zwchow92@126.com (Z.-W.Z.); m19914874966@163.com (Y.-Q.L.); t2932516179@163.com (L.-Q.T.); 2Key Laboratory of Tea Science in Fujian Province, College of Horticulture, Fujian Agriculture and Forestry University, Fuzhou 350002, China; doris1831036881@126.com (Q.-Y.W.); wy_forward@163.com (Y.W.); dengtt0927@fafu.edu.cn (T.-T.D.); 15860687168@163.com (J.-H.H.)

**Keywords:** black tea, LOX pathway, volatile, tea processing

## Abstract

Volatile fatty acid derivatives (VFADs) play a significant role in contributing to flowery–fruity flavor black tea. Innovative black tea is typically crafted from aroma-intensive tea cultivars, such as Jinmudan, using defined production methodologies. In this study, the during-processing tea leaves of innovative black tea were applied as materials, and we selected a total of 45 VFADs, comprising 11 derived aldehydes, nine derived alcohols, and 25 derived esters. Furthermore, the dynamic variations of these VFADs were uncovered. Transcriptome analysis was performed to identify genes involved in the LOX (lipoxygenase) pathway, resulting in the identification of 17 *CsLOX* genes, one hydrogen peroxide lyase (*CsHPL*) gene, 11 alcohol dehydrogenases (*CsADH*) genes, 11 genes as acyl CoA oxidase (*CsACOX*) genes, and three allene oxide synthase (*CsAOS*) genes. Additionally, the expression levels of these genes were measured, indicating that the processing treatments of innovative black tea, particularly turn-over and fermentation, had a stimulation effect on most genes. Finally, qRT-PCR verification and correlation analysis were conducted to explain the relationship between VFADs and candidate genes. This study aims to provide a reference for illuminating the formation mechanisms of aroma compounds in innovative black tea, thereby inspiring the optimization of innovative processing techniques and enhancing the overall quality of black tea.

## 1. Introduction

Among the six traditionally recognized tea categories, black tea obtains high reputation due to its sweet and mellow taste coupled with a long-lasting and abundant aroma [1,2]. The key steps in the primary processing of traditional black tea encompass withering, rolling, fermentation, and drying [3]. The fresh leaves plucked from tea cultivars ‘Fuding Dabaicha’, ‘Fu’an Dabaicha’, and locally sexual tea cultivar, are typically employed as the raw materials for traditional Congou black tea production. This black tea is distinguished by its unique aroma profile, which is characterized by notes of sweetness, honey, and caramel [4]. Previous studies had revealed the aroma-active compounds in traditional Congou black tea. For example, Gao et al. [5] have identified 1-pentanol, propyl hexanoate, and linalool as volatile metabolites contributing to the sweet and floral aroma character in traditional Tanyang Congou black tea. Wang et al. [6] have conducted a study on the essential flavor compounds that comprise the ‘Keemun aroma’ in Qihong Congou black tea. Concurrently, the distinctive volatile compounds found in regional traditional Congou black tea like Zunyi [7], Chuanhong [8], and Xinyang [9] Congou black tea were also reported.

Not only with respect to tea cultivar, but the manufacturing process is also crucial in the formation of tea quality. In the black tea post-harvest procedure, the fermentation process (color conversion) plays a decisive role in determining the quality character. Other processing methods like withering [10], rolling [11], and drying [12] also lay a foundation for this important character. Therefore, adjustments with respect to processing were applied to improve the quality of Congou black tea, and innovative black tea was produced [13].

Firstly, the innovative black tea is produced using aroma-enriched oolong tea varieties, particularly ‘Jinmudan’, ‘Zimeigui’, and ‘Mingke No. 1′. Secondly, in contrast to traditional black tea production methods, the innovative black tea incorporates the turn-over step during withering, which originated from oolong tea processing [14,15]. These adjustments have led the innovative black tea possessing a floral and fruity aroma, amber-colored liquor, and a sweet taste, which have won widespread consumer acceptance in recent times. Zhou et al. [16] investigated the aroma and flavor profiles of four grades of innovative black tea standard samples, developing aroma wheels tailored to innovative black tea grade distinctions. Wu et al. [17] identified indole, methyl salicylate, and δ-decalactone as characteristic aroma compounds within innovative black tea. It was reported that turn-over treatment can effectively stimulate the conversion of aroma compounds, resulting in a higher floral and fruity aroma quality [18,19] and thereby significantly enhancing its market value, like Hunan black tea [20] and summer black tea [21].

Fatty acids serve as aroma precursors in tea leaves, transforming into volatile fatty acid derivatives (VFADs) under biotic or abiotic stresses [22,23]. Biotic and abiotic stresses, including herbivore attacks and processing treatments, could regulate the genes involved in the fatty acid pathway to elicit a response [24]. *α*-Linolenic acid and linoleic acid are typical precursors of VFADs [18]. The activation of the lipoxygenase (LOX) pathway, located in the chloroplasts of green leaves in the plant kingdom, characterizes the response of tea leaves to various abiotic stresses during post-harvest processing. Previous research demonstrated that LOX and its associated genes actively respond to mechanical damage induced by turn-over treatment, facilitating PUFAs oxidative degradation [25,26,27]. The oxidative products of PUFAs undergo further cleavage mediated by hydroperoxide lyase (HPL). HPL-associated genes exhibit synergistic responses to biotic and abiotic stress during the post-harvest processing of tea leaves [28]. Aldehydes, generated by HPL, are subsequently reduced to alcohols via the action of alcohol dehydrogenase (ADH), which is a versatile zinc-containing metalloenzyme. Gao et al. [29] observed that the *CsADH2* gene played an important role in the formation of VFADs during the post-harvest processing of white tea. Additionally, Liu et al. [30] demonstrated the pivotal role of ADH activation in the transformation of C6 aroma compounds in tea plants. Apart from the LOX–HPL pathway, solar withering, and turnover processes, which occur concurrently with water deficit, heat, light, and wounding stress, stimulate the activation of the LOX–AOS pathway. This pathway facilitates the transcription and expression of key structural genes, such as *CsAOSs* and *CsACOXs*, thereby enhancing the production of volatile metabolites, notably methyl jasmonate, which positively contributes to the development of tea aroma [31,32] (Figure 1). In summary, turn-over serves as an efficacious approach to stimulate the formation of VFADs during the post-harvest processing of tea leaves, with VFADs playing a pivotal role in aroma generation during tea production. However, it still needs to be revealed that the profile and dynamic change of VFADs and the candidate genes within the LOX pathway and their expression, as well as regulation during innovative black tea processing procedure.

In this study, VFADs were selected first, and their dynamic changes were explicated during innovative black tea processing. Secondly, a transcriptome analysis was performed to identify the genes related to the LOX pathway, and subsequently, the expression levels of these genes were measured. Lastly, qRT-PCR verification and correlation analysis were conducted to illuminate the relationship between VFADs and candidate genes. In conclusion, all the preliminary results threw light on the nature of the formation pattern and molecular mechanism of VFADs related to the floral and fruity aromas of innovative black tea in order to better understand the effect of processing factors on the aroma quality in black tea, thereby providing valuable insights for the targeted manufacturing and quality enhancement of black tea.

## 2. Materials and Methods

### 2.1. Tea Material

The fresh tea leaves (*Camellia sinensis* cv. Jinmudan) all were plucked at the tea garden of Fujian Xianliangge Tea Ltd. (Fu’an, China) on October 2022. All leaves were collected from the same branch with the standard “one bud and three leaves” approach, which is typical for the process industrially [33]. Five processing steps were selected to collect materials for volatile compounds detection and RNA sequencing, marked as fresh leaves (L), after solar withering (W), after second time turn-over (T), after second time spreading (S), after fermentation (F) (Figure 2). Tea leaves were collected and then deep frozen in liquid nitrogen, and they were maintained at −80 °C for further study.

### 2.2. Detection of Fatty Acid Derived Compounds

#### 2.2.1. Sample Preparation and Treatment

Materials samples were ground to a powder in liquid nitrogen. A total of 500 mg (1 mL) of the powder was transferred immediately to a 20 mL headspace vial (Agilent, Palo Alto, CA, USA), containing NaCl saturated solution, to inhibit any enzyme reaction. The vials were sealed using crimp top caps with TFE-silicone headspace septa (Agilent, Palo Alto, CA, USA). At the time of SPME analysis, each vial was placed in 60 °C for 5 min; then, a 120 µm DVB/CWR/PDMS fiber (Agilent, Palo Alto, CA, USA) was exposed to the headspace of the sample for 15 min at 60 °C.

#### 2.2.2. GC-MS Conditions

After sampling, desorption of the VOCs (volatile compounds) from the fiber coating was carried out in the injection port of the GC apparatus (Model 8890; Agilent) at 250 °C for 5 min in the splitless mode. The identification and quantification of VOCs were carried out using an Agilent Model 8890 GC and a 7000D mass spectrometer (Agilent) equipped with a 30 m × 0.25 mm × 0.25 μm DB-5MS (5% phenyl-polymethylsiloxane) capillary column. Helium was used as the carrier gas at a linear velocity of 1.2 mL/min. The injector temperature was kept at 250 °C and the detector at 280 °C. The oven temperature was programmed from 40 °C (3.5 min), increasing at 10 °C/min to 100 °C at 7 °C/min to 180 °C, at 25 °C/min to 280 °C, hold for 5 min. Mass spectra were recorded in electron impact (EI) ionization mode at 70 eV. The quadrupole mass detector, ion source and transfer line temperatures were set, respectively, at 150, 230, and 280 °C. The MS was selected ion monitoring (SIM) mode was used for the identification and quantification of analysis.

MassHunter tool (Agilent) was used to process mass spectrometry files and select quantitative ions for integration and calibration processing. The retention index (RI) range was defined between (RI target − threshold) and (RI target + threshold), and the threshold was set as 60. All the detected VFADs in LOX–HPL branch within the RI range were selected as candidates.

### 2.3. RNA Isolation and Sequencing

Total RNA was isolated, and its concentration and purity were detected by Nanodrop 2000. The integrity of RNA and RIN values was determined by Agilent 2100. Libraries were built according to the requirements of a total RNA volume of ≥1 ug and a concentration of ≥35 ng/μL, OD260/280 ≥ 1.8, OD260/230 ≥ 1.0. Clean reads were obtained after removing low-quality reads or reads that contained adaptors. Then, the ‘Huangdan’ tea plant genome was introduced to calculate gene alignment using Hisat v2.1.0. Feature Counts v1.6.2, following which the FPKM of each gene based on the gene length was obtained.

### 2.4. Quantitative Real-Time PCR Assay

Quantitative real-time PCR (qRT-PCR) analysis was conducted to verify the RNA sequencing results. cDNA was synthesized by using PrimeScript RT Reagent Kit with a gDNA Eraser (TaKaRa, Dalian, China), and qRT-PCR was performed based on this procedure: 37 °C for 2 min, 95 °C for 3 min, and 95 °C 5 s continuing 40 circles; 60 °C for 30 s, finally dissociation stage. *CsGADPH* was selected as the reference gene to normalize the data [34,35], and the primer used was designed by DNAMAN (Appendix A). The relative expression level of genes was calculated using the 2^−∆∆Ct^ method [36].

### 2.5. Data Processing and Statistical Analysis 

Statistical analysis and figures drawing were conveyed by WPS 20.0 software. The line chart was performed using the OmicStudio tools at https://www.omicstudio.cn/tool (accessed on 29 July 2024) [37]. The bar chart was performed using the Prism (GraphPad, Version 6.01, GraphPad Software Inc., San Diego, CA, USA). All experimental data are presented as mean ± standard deviation. Differences in the content of VFADs were determined using Tukey’s honest significant difference (HSD) test.

## 3. Results and Discussion

### 3.1. The Profile and Dynamic Change of Volatile Fatty Acid Derived Compounds

A total of 45 types of VFADs were identified, consisting of 11 aldehyde derivatives, nine alcohol derivatives, and 25 ester derivatives (Appendix A). The dynamic changes of these VFADs were illuminated during the innovative black tea production process.

The identified derived aldehydes included 6-nonenal, (*Z*)-7-decen-1-al, (*E*)-2-decenal, (*E*)-4-decenal, (*Z*)-4-heptenal, tridecanal, (*E*,*E*)-2,4-nonadienal, (*Z*)-2-decenal, hexanal, (*2E*,*4Z*)-2,4-decadienal, and (*E*,*E*)-2,4-undecadienal, as illustrated in Figure 3. The varying trends of these eleven compounds can be broadly classified into two categories: those that exhibited an initial decrease followed by an increase and those that displayed a gradual decrease or increase. Six compounds—(*E*)-6-nonenal, (*Z*)-7-decen-1-al, (*E*)-2-decenal, (*Z*)-2-decenal, (*E*,*E*)-2,4-nonadienal, and (*2E*,*4Z*)-2,4-decadienal—exhibited predominant peaks in fresh leaves (L), with the exception of (*E*,*E*)-2,4-nonadienal and (*2E*,*4Z*)-2,4-decadienal, which reached their maxima during the fermentation step (F) at levels of 0.0084 μg/g and 0.0016 μg/g, respectively. These six compounds exhibited their minimum concentrations during the second turnover (T) or second spreading (S) stages. This variation can be attributed to mechanical stress induced by turnover and water deficit stress resulting from spreading; both factors facilitated the emission and transformation of aldehydes. Only (*E*)-4-decenal exhibited a gradual decrease throughout the processing procedure, reaching its minimum concentration during the fermentation (F) stage at 0.0101 μg/g. In contrast, four compounds—(*Z*)-4-heptenal, hexanal, tridecanal, and (*E*,*E*)-2,4-undecadienal—demonstrated a gradual increase and peaked during fermentation (F), indicating a pronounced response to the abiotic stresses induced by processing. The difference analysis was conveyed, and the results are listed with lowercase letters; it can be seen that most of the derived compounds observed significant differences except (*Z*)-4-heptenal, hexanal, tridecanal, and (*E*,*E*)-2,4-undecadienal. Among them, there existed significant differences (*p* < 0.05) between fermentation (F) and fresh leaves (L) in compounds (*Z*)-4-heptenal, tridecanal, and (*E*,*E*)-2,4-undecadienal, while in the hexanal, there existed a significant difference (*p* < 0.05) in two random treatments during innovative black tea processing. 

A total of nine derived alcohols were identified: (*Z*)-2-penten-1-ol, (*E*)-3-nonen-1-ol, (*E*)-2-decen-1-ol, 1-decanol, 2,4-decadien-1-ol, n-tridecan-1-ol, 1-octanol, 1-nonen-4-ol, and (*E*)-2-nonen-1-ol (Figure 4). These compounds exhibited varying degrees of fluctuation and reached their peak concentrations during fermentation (F), with (*Z*)-2-penten-1-ol showing the highest content in the fermented treatment at 0.4360 μg/g. Certain compounds, such as 1-decanol, 2,4-decadien-1-ol, and 1-octanol, demonstrated slight increases during the second turnover phase (T), with values of 0.0300 μg/g, 0.1908 μg/g, and 0.2672 μg/g, respectively. Notably, 1-nonen-4-ol peaked at its maximum level of 0.1569 μg/g during this stage, suggesting that the turnover treatment may enhance the accumulation of derived alcohols. Additionally, (*E*)-2-nonen-1-ol exhibited significant fluctuations—particularly a sharp decline from fresh leaves (L) to withering (W)—indicating that sudden water stress might have contributed to its degradation. Nevertheless, (*E*)-2-nonen-1-ol gradually re-formed and accumulated over time upon the subsequent treatment. The difference analysis revealed that only significant differences existed for compound levels between treatments for 1-nonen-4-ol. Other compounds like 1-decanol and 1-octanol both demonstrated significant differences between fermentation and other treatments. As for *(E)*-2-nonen-1-ol, this compound also showed significant differences when compared to other treatments.

A total of 25 types of derived esters were identified, representing the majority and displaying diverse trends categorized into three distinct groups (Figure 5). The first category comprised five compounds that exhibited a fluctuating decrease: hexanoic acid 1-methylethyl ester, (*Z*)-4-decenoic acid methyl ester, formic acid octyl ester, acetic acid octyl ester, and n-amyl isovalerate. The maximum concentrations of these compounds were detected in fresh leaves (L), followed by a decline during the processing procedure. Notably, the level of acetic acid octyl ester peaked during the second turnover (T) stage at 0.0381 μg/g before declining; conversely, it increased after this stage to reach 0.0048 μg/g, suggesting that mechanical stress may promote the formation of these two compounds. The second group included six compounds: butanoic acid 2-methyl-2-methylpropyl ester, nonanoic acid ethyl ester, heptanoic acid ethyl ester, hexanoic acid 3-hexenyl ester, octanoic acid methyl ester, and isobutyl isovalerate. All compounds reached their peak levels during the second turnover (T) stage before subsequently declining; however, nonanoic acid ethyl ester peaked during the follow-up spreading (S) phase at 0.6962 μg/g. Mechanical stress induced by turning over facilitated the formation of these six compounds. Nevertheless, water deficit due to spreading and heat generated in fermentation hindered their accumulation. The latter stages yielded significant amounts of major esters derivatives, which exhibited gradual increases with slight fluctuations. Among these, hexanoic acid butyl ester, (*E*)-butanoic acid 3-hexenyl ester, butanoic acid 5-hexenyl ester, and hexanoic acid methyl ester peaked during the second turnover (T) before declining slightly. Similarly, hexanoic acid 5-hexenyl ester and 2-butenoic acid hexyl ester exhibited their peak concentrations during the spreading (S) phase (0.0048 μg/g and 0.0983 μg/g, respectively), followed by a subsequent decline. This observation indicates that the abiotic stresses induced by turning over and spreading have distinct effects on the accumulation of various compounds. In contrast, other compounds such as ethyl 5-methylhexanoate, butanoic acid octyl ester, and butanoic acid 3-methylbutyl ester progressively reached their maximum levels during fermentation (F) treatment. This finding suggest that these compounds positively responded to the entire innovative black tea processing regimen. Notably, ethyl 5-methylhexanoate, butanoic acid,octyl ester, butanoic acid, 3-methylbutyl ester, and eight additional compounds (Figure 5R–Y) demonstrated peak concentrations during fermentation (F) and exhibited significant differences (*p* < 0.05) when compared to other treatments. The differential analysis revealed substantial variations among most compounds across different treatments, indicating that the innovative black tea processing methods significantly influenced the trends in derived ester compound accumulation.

### 3.2. Evaluation of Total RNA Quality and the Measurement of Differential Genes

Five samples from the processing of innovative black tea were chosen as the subjects of our research, and the Nanodrop 2000 was used to measure the concentration and purity of the extracted RNA. Agarose gel electrophoresis was employed to assess the integrity of the RNA. The results of these tests are presented in Appendix A. For library construction, the total RNA content had to exceed 1 μg, with a concentration greater than 35 ng/μL, an OD260/280 ratio greater than 1.8, and an OD260/230 ratio exceeding 1.0. The purity, concentration, and integrity of the extracted RNA all met the criteria necessary for library construction, indicating that it could be utilized in subsequent experiments.

Principal Component Analysis (PCA) is a multivariate statistical technique that extracts important variables via the linear transformation of numerous variables, enabling the classification of comprehensive indicators according to predefined rules. Our study demonstrated that PCA effectively distinguished five samples, with the first principal component accounting for 58.44% and the second for 12.14%, indicating the reliability of this analytical approach. Specifically, fresh leaves (L) were positioned in the first quadrant, signifying their initial state prior to post-treatment. Withering leaves (W) and second turnover samples (T) resided in the third and fourth quadrants, respectively, which is indicative of water deficit and mechanical stress experienced during their respective treatments. The subsequent stages of spreading (S) and fermentation (F) were situated in the second quadrant. The close proximity between duplicate samples indicates the high repeatability of this analytical mode (Figure 6).

The number of differentially expressed genes gradually increased throughout the processing procedure, peaking during fermentation (F, 14 026). Furthermore, when comparing the spreading (S) and fermentation (F) stages, the number of differentially expressed genes was lower. Consequently, it is hypothesized that water deficit and mechanical stress during early stages, such as withering and turning, facilitated the upregulation of numerous genes. However, as spreading and fermentation progressed, the rate of differentially expressed gene expression declined. Both these findings suggest that early processing treatments maintained tea leaf activity and significantly altered gene expression, profoundly influencing physiological and metabolic processes and ultimately contributing to the flavor and quality of innovative black tea (Figure 7).

### 3.3. The Selection and Expression Analysis of Fatty Acid Pathway Related Genes

Fatty acid metabolism serves as a crucial source of aroma compounds in the tea plant. The precursors of fatty acids undergo decomposition, yielding VFADs that contribute to green leaf, floral, and even fruity aroma profiles via biological and physical processes. Transcriptome screening revealed a total of 416 genes involved in the fatty acid metabolism pathway. In the LOX–HPL branch pathway, 17 genes encoding LOX, 1 gene encoding HPL, and 10 genes encoding ADH were identified. Alcohol acyltransferase (AAT) was absent from the identified genes. In the *β*-oxidation branch pathway, 11 genes encoding ACOX were identified, which act as rate-limiting enzymes for the initial step of *β*-oxidation and participate in lactone formation. Within the jasmonic acid branch pathway, three genes encoding AOS were identified. 

Differentially expressed genes were selected, and their expression levels were analyzed, as depicted in Figure 8. LOX has the ability to convert unsaturated fatty acids into 9/13 hydroperoxides. In this study, the expression levels of *CsLOXs* were elevated during the withering (W) and turnover (T) stages of innovative black tea processing, whereas during the fermentation stage, their expression remained relatively stable. The 9/13 hydroperoxide served as an unstable intermediate in the biochemical pathway, readily undergoing conversion into C6/C9 aldehydes catalyzed by HPL. In this study, *CsHPL* exhibited a robust response during the withering (W) and spreading (S) stages. 

LOX played a crucial role in converting *α*-linolenic acid/linolenic acid to C6 aldehydes and C6 alcohols. Furthermore, LOX exhibits distinct catalytic properties; some isoforms exclusively catalyze the oxidation of triglycerides, while others specialize in fatty acid oxidation [38,39]. Specifically, LOX selectively oxidizes a limited set of polyunsaturated fatty acids with (*Z*)-pentacene structures, such as linoleic and linolenic acids [40], which are abundant in fresh ‘Jinmudan’ tea leaves. Consequently, the abundant fatty acid precursors in the ‘Jinmudan’ tea leaves serve as ample substrates for LOX-mediated reactions. Apart from these, LOX exhibits robust oxidative capabilities and can participate in reactions under both aerobic and anaerobic conditions. The enzymatic oxidation of LOX under anaerobic conditions is more intricate compared to aerobic reactions. For example, during anaerobic oxidation of linoleic acid, unreacted linoleic acid forms a series of dimers alongside hydroperoxides [41]. Prior research has focused on comparing changes in VFADs and regulatory factors in fatty acid metabolism pathways under both aerobic and anaerobic processing conditions [28]. The results indicate that anaerobic conditions favor enzymatic reduction of C6 aldehydes, leading to the formation and accumulation of C6 alcohols and derived esters, whereas aerobic conditions promote the nonenzymatic oxidation of C6 aldehydes, resulting in the accumulation of C6 acids and derived esters. Under anaerobic conditions, the accumulation of volatile compounds in processed tea increased in response to exogenous abiotic stress resulting from hypoxia, such as the accumulation of CO_2_ and water [42]. Based on these findings, during innovative black tea processing, compared to withering and spreading methods, the turnover and fermentation treatments weakened oxygen circulation, and increased humidity and temperature, potentially inhibiting the expression of *CsLOX5* and *CsHPL*.

The initial engagement of AOS facilitates the progression of 9/13 hydroperoxide toward jasmonic acid biosynthesis. In this study, *CsAOSs* demonstrated high expression levels during the later processing stages. Subsequently, C6/C9 aldehydes underwent further conversion into C6/C9 alcohols via the catalysis of ADH. In this study, the majority of *CsADHs* responded favorably to the processing of innovative black tea, with their expression levels gradually escalating. Nevertheless, two *CsADHs* displayed peak expression levels in fresh leaves (L), which subsequently decreased during the processing stages. Unfortunately, we did not identify the AAT regulating the esters of C6/C9 alcohols. However, the first rate-limiting enzyme ACOX in the *β*-oxidation pathway of fatty acids was found to be highly responsive to the processing of innovative black tea. Overall, the processing treatments of innovative black tea promoted the expression of the genes involved in fatty acid metabolism, with some genes reaching peak expression levels during the oxidation and fermentation stages.

The *β*-oxidation reaction, alongside the LOX–HPL pathway, serves as a crucial branch for the degradation and metabolism of fatty acids. Additionally, the formation of aroma compounds derived from fatty acids in tea necessitates the involvement of this oxidative branch. For instance, the synthesis of cyclic-fatty-acid-derived compounds necessitates LOX oxidation to generate hydrogen peroxide, which subsequently undergoes catalysis by AOS and AOC (allene oxide cyclase), yielding OPDA (oxo phytofatty acid). OPDA undergoes reduction followed by a three-step *β*-oxidation reaction, leading to the formation of jasmonic acid, which is a precursor of numerous jasmonic acid derivatives such as methyl jasmonate, methyl dihydrojasmonate, and jasmine lactone [43,44]. In this study, *CsAOS* exhibited diverse responses to abiotic stresses induced by the innovative black tea processing method. 

### 3.4. The qRT-PCR Verification of Selected Fatty Acid Metabolic Pathway Genes

Fourteen candidate genes associated with the fatty acid metabolism pathway were selected for RT-qPCR validation based on their expression levels (greater than 10.0) and fold changes (greater than 2.0). A bar chart displayed the FPKM values, while a line chart showed the RT-qPCR validation results for each gene. The consistent values and trends suggest that the transcriptome data are reliable and accurately reflect the expression levels of he key genes involved in fatty acid metabolism during innovative black tea processing (Appendix A).

Five genes, *CsLOX1* (TGY077916), *CsLOX2* (TGY094953), *CsLOX3* (TGY024305), *CsLOX4* (TGY017079), and *CsLOX5* (TGY094955), were selected. Only one gene, *CsHPL* (TGY044402), was chosen. Based on the FPKM values of *CsLOXs* and *CsHPL*, it was evident that all genes responded to mechanical damage and water deficit during the production of innovative black tea. Mechanical stress was the primary factor, with some genes showing higher expression levels during fermentation (F). Four genes, *CsAOS1* (TGY084375), *CsAOS2* (TGY084373), *CsAOS3* (TGY021554), and *CsAOS4* (TGY031841), were selected. The expression levels of these genes varied, with mechanical stress, water deficit, and heat stress all inducing high expression of *CsAOSs*. Additionally, two genes, *CsADH1* (TGY037356) and *CsADH2* (TGY097376), and two genes, *CsACOX1* (TGY062748) and *CsACOX2* (TGY007218), were chosen. *CsADHs* exhibited a positive response to water deficit during spreading (S), while *CsACOXs* responded positively to water deficit but less significantly to mechanical stress during turnover (T).

In summary, the RT-qPCR results concurred with the FPKM values, suggesting that these outcomes can accurately portray the diverse responses of genes to the diverse stresses encountered during the innovative black tea process.

### 3.5. The Correlation Analysis between Fatty Acid Derived Compounds and Selected Genes

A Pearson correlation analysis was conducted between the selected 45 kinds of VAFDs and 14 related genes, as shown in Figure 9. It could be roughly divided and clustered into three groups. The first group contained five genes that had closely positive correlation with most derived compounds, which are listed as *CsAOS1*, *CsAOS3*, *CsLOX3*, *CsADH2*, and *CsAOS2*. Among them, *CsAOS3* had extremely significant correlation with derived compounds like butanoic acid,octyl ester, butyl caprylate, ethyl 9-decenoate, and (*E*,*E*)-2,4-undecadienal. The second group contained four genes, including *CsLOX1*, *CsADH1*, *CsLOX5*, and *CsLOX2*, that negatively related to the majority of derived compounds except heptanoic acid,ethyl ester, butanoic acid,2-methyl-,2-methylpropyl ester, and isobutyl isovalerate (positively related). The last group contained five genes named *CsAOS4*, *CsLOX4*, *CsACOX2*, *CsACOX1*, and *CsHPL*, which just only related to most derived compounds. Among them, *CsAOS4* and *CsLOX4* both had extremely significant correlation with butanoic acid,2-methyl-,2-methylpropyl ester and isobutyl isovalerate, and they had positive correlations with octanoic acid and methyl ester. In summary, the selected 14 fatty acid related genes had convincing correlations with most VFADs. 

These were broadly categorized into three distinct groups. The first group comprised five genes—*CsAOS3*, *CsAOS1*, *CsLOX3*, *CsADH2*, and *CsAOS2*—which exhibited strong positive correlations with most derived compounds. Specifically, *CsAOS3* demonstrated highly significant correlations with derived compounds such as butanoic acid, octyl ester, butyl caprylate, ethyl 9-decenoate, and (*E*,*E*)-2,4-undecadienal. The second group consisted of *CsLOX1*, *CsADH1*, *CsLOX5*, and *CsLOX2*, which displayed negative correlations with most VFADs, except for heptanoic acid ethyl ester, butanoic acid 2-methyl-, 2-methylpropyl ester, and isobutyl isovalerate (which were positively correlated). The third group encompassed *CsAOS4*, *CsLOX4*, *CsACOX2*, *CsACOX1*, and *CsHPL*, which were associated with most VFADs. Notably, *CsAOS4* and *CsLOX4* exhibited highly significant correlations with butanoic acid 2-methyl-, 2-methylpropyl ester, and isobutyl isovalerate, and they also showed positive correlations with octanoic acid methyl ester.

*β*-Oxidation is also implicated in the biosynthesis of lactone compounds. Initially, fatty acids are oxidized by LOX to generate hydroperoxides, which subsequently undergo *β*-oxidation, initiating the lactone synthesis pathway. This process culminates in the production of acyl-CoA, which is an essential acyl donor in ester compound synthesis [45,46]. Subsequently, multiple reactions take place to synthesize lactone compounds, with ACOX serving as a crucial rate-limiting enzyme [47]. Nevertheless, research into the function and analysis of ACOX during the post-harvest stage of tea remains scarce. Consequently, future research should endeavor to explore the regulatory role of ACOX on lactone compounds, contributing to a more comprehensive understanding of this process.

## 4. Conclusions

In summary, this study examined a total of 45 VFADs, comprising 11 derived aldehydes, nine derived alcohols, and 25 derived esters, based on LOX–HPL branch and transcriptome sequencing at five processing points of innovative black tea, shedding light on the variation patterns of VFADs at the key processing treatments of innovative black tea, subsequently exploring and analyzing the key genes associated with the fatty acid metabolism pathway and its related branches. Finally, RT-qPCR validation and correlation analysis were performed. The study results identified a total of 17 *CsLOX* genes, one *CsHPL* gene, 10 *CsADH* genes, 11 *CsACOX* genes, and three *CsAOS* genes, while alcohol acyltransferase (AAT) was not detected. However, specific processing stresses can exert inhibitory influences on particular genes, resulting in reduced expression levels. RT-PCR validation confirmed a general concordance between FPKM values in the transcriptome and relative expression, signifying the reliability of transcriptome data in reflecting the expression profiles of key fatty acid metabolism genes during the processing of innovative black tea.

## Figures and Tables

**Figure 1 foods-13-03108-f001:**
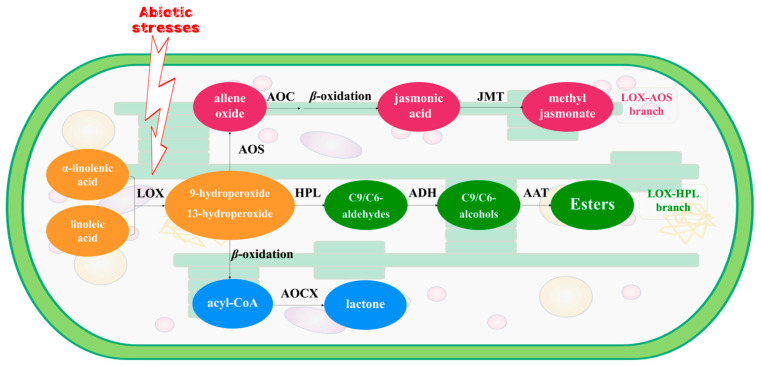
The LOX pathway of *α*-linolenic acid and linoleic acid located in chloroplasts of tea leaf. LOX: lipoxygenase; AOS: allene oxide synthase; AOC: allene oxide cyclase; JMT: jasmonic acid carboxyl methyltransferase; HPL: hydroperoxide lyase; ADH: alcohol dehydrogenase; AAT: alcohol acyltransferases; ACOX: acyl-CoA oxidase.

**Figure 2 foods-13-03108-f002:**
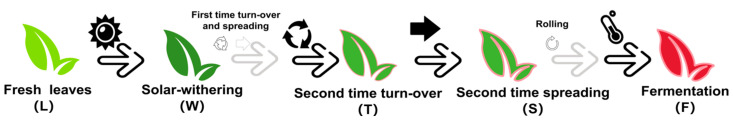
The processing steps of innovative black tea used as materials.

**Figure 3 foods-13-03108-f003:**
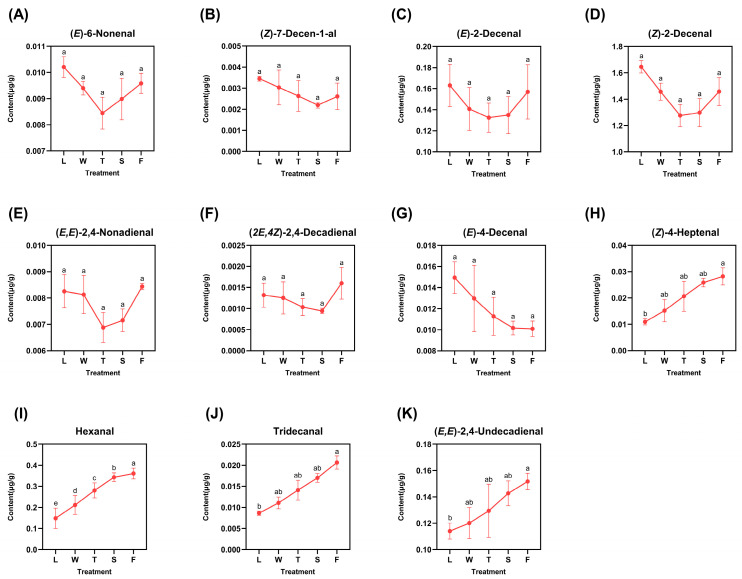
The dynamic change of fatty-acid-derived aldehyde compounds. L: fresh leaves; W: after solar withering; T: after second time turnover; S: after second time spreading; F: after fermentation. Note: Different lowercase letters (a, b, c, d, e) represent significant differences at *p* < 0.05. (**A**): (*E*)-6-Nonenal. (**B**): (*Z*)-7-Decen-1-al. (**C**): (*E*)-2-Decenal. (**D**): (*Z*)-2-Decenal. (**E**): (*E*,*E*)-2,4-Nonadienal. (**F**): (*2E*,*4Z*)-2,4-Decadienal. (**G**): (*E*)-4-Decenal. (**H**): (*Z*)-4-Heptenal. (**I**): Hexanal. (**J**): Tridecanal. (**K**): (*E*,*E*)-2,4-Undecadienal.

**Figure 4 foods-13-03108-f004:**
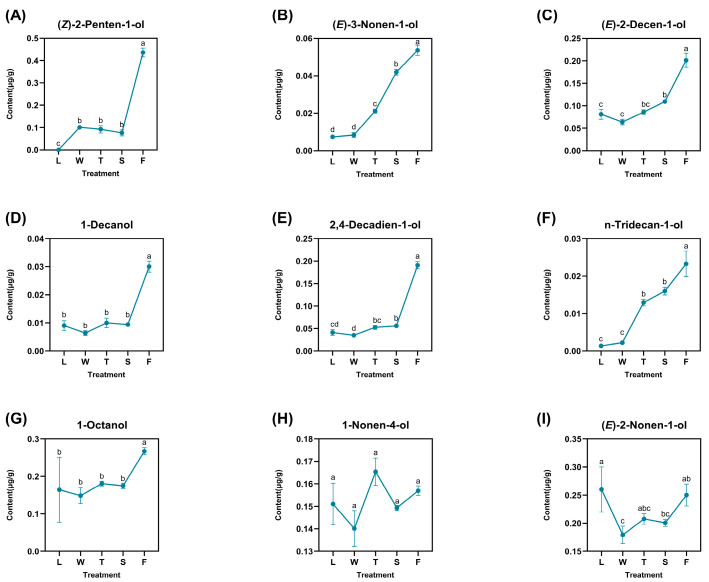
The dynamic change of fatty-acid-derived alcohol compounds. L: fresh leaves; W: after solar withering; T: after second time turnover; S: after second time spreading; F: after fermentation. Note: Different lowercase letters (a, b, c, d) represent significant differences at *p* < 0.05. (**A**): (*Z*)-2-Penten-1-ol. (**B**): (*E*)-3-Nonen-1-ol. (**C**): (*E*)-2-Decen-1-ol. (**D**): 1-Decanol. (**E**): 2,4-Decadien-1-ol. (**F**): n-Tridecan-1-ol. (**G**): 1-Octanol. (**H**): 1-Nonen-4-ol. (**I**): (*E*)-2-Nonen-1-ol.

**Figure 5 foods-13-03108-f005:**
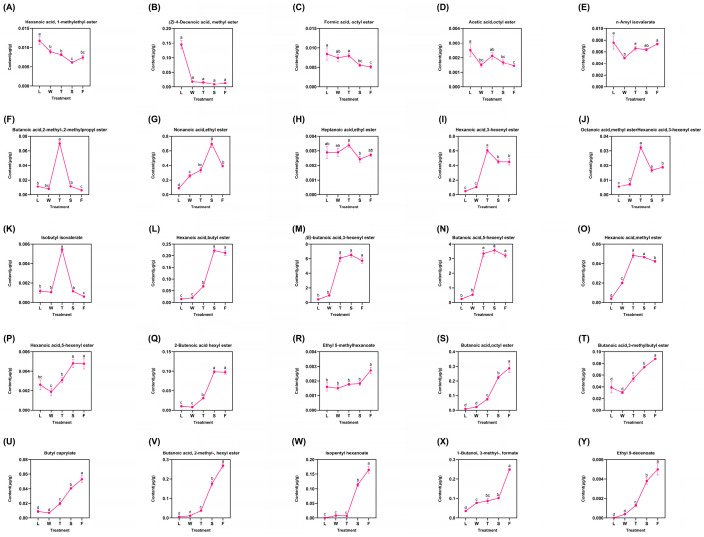
The dynamic change of fatty-acid-derived ester compounds. L: fresh leaves; W: after solar withering; T: after second time turnover; S: after second time spreading; F: after fermentation. Note: Different lowercase letters (a, b, c, d) represent significant differences at *p* < 0.05. (**A**): Hexanoic acid, 1-methylethyl ester. (**B**): (*Z*)-4-Decenoic acid, methyl ester. (**C**): Formic acid, octyl ester. (**D**): Acetic acid, octyl ester. (**E**): n-Amyl isovalerate. (**F**): Butanoic acid,2-methyl-, 2-methylpropyl ester. (**G**): Nonanoic acid, ethyl ester. (**H**): Heptanoic acid, ethyl ester. (**I**): Hexanoic acid, 3-hexenyl ester. (**J**): Octanoic acid, methyl ester. (**K**): Isobutyl isovalerate. (**L**): Hexanoic acid, butyl ester. (**M**): (*E*)-butanoic acid, 3-hexenyl ester. (**N**): Butanoic acid, 5-hexenyl ester. (**O**): Hexanoic acid, methyl ester. (**P**): Hexanoic acid, 5-hexenyl ester. (**Q**): 2-Butenoic acid hexyl ester. (**R**): Ethyl 5-methylhexanoate. (**S**): Butanoic acid, octyl ester. (**T**): Butanoic acid, 3-methylbutyl ester. (**U**): Butyl caprylate. (**V**): Butanoic acid, 2-methyl-, hexyl ester. (**W**): Isopentyl hexanoate. (**X**): 1-Butanol, 3-methyl-, formate. (**Y**): Ethyl 9-decenoate.

**Figure 6 foods-13-03108-f006:**
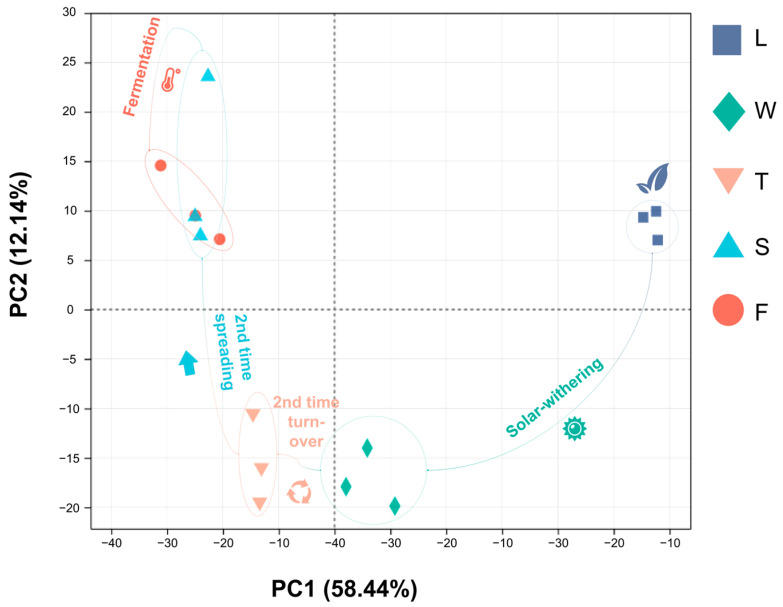
The PCA analysis of different processing innovative black tea samples. L: fresh leaves; W: after solar withering; T: after second time turnover; S: after second time spreading; F: after fermentation. PC1: principal component 1; PC2: principal component 2.

**Figure 7 foods-13-03108-f007:**
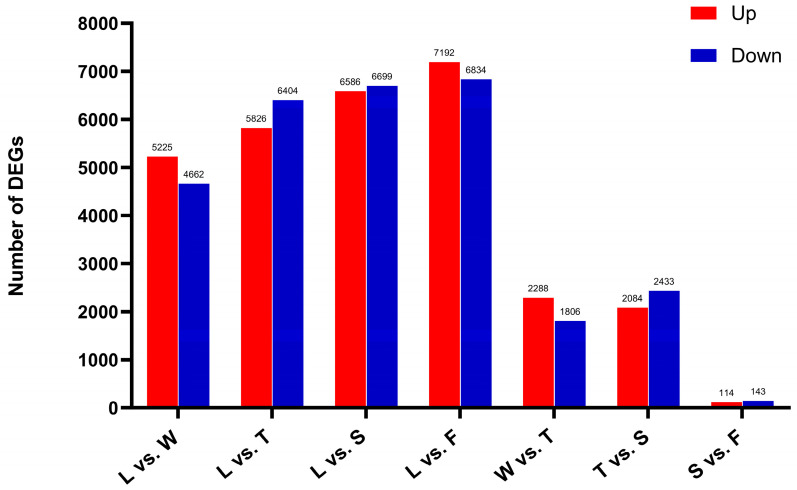
The differentially expressed genes in different comparison groups. L: fresh leaves; W: after solar withering; T: after second time turnover; S: after second time spreading; F: after fermentation. DEGs: Differentially expressed genes.

**Figure 8 foods-13-03108-f008:**
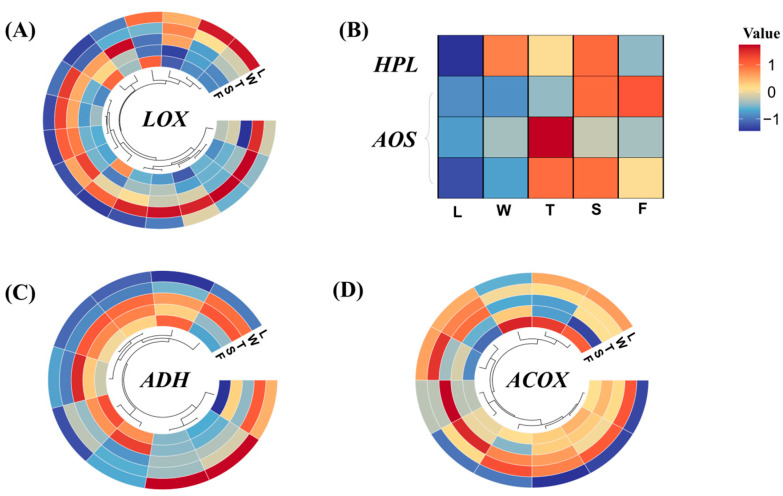
The expression of genes in fatty acid metabolism and its related branch pathway. L: fresh leaves; W: after solar withering; T: after second time turnover; S: after second time spreading; F: after fermentation. LOX: lipoxygenase; HPL: hydroperoxide lyase; ADH: alcohol dehydrogenase; ACOX: acyl-CoA oxidase.

**Figure 9 foods-13-03108-f009:**
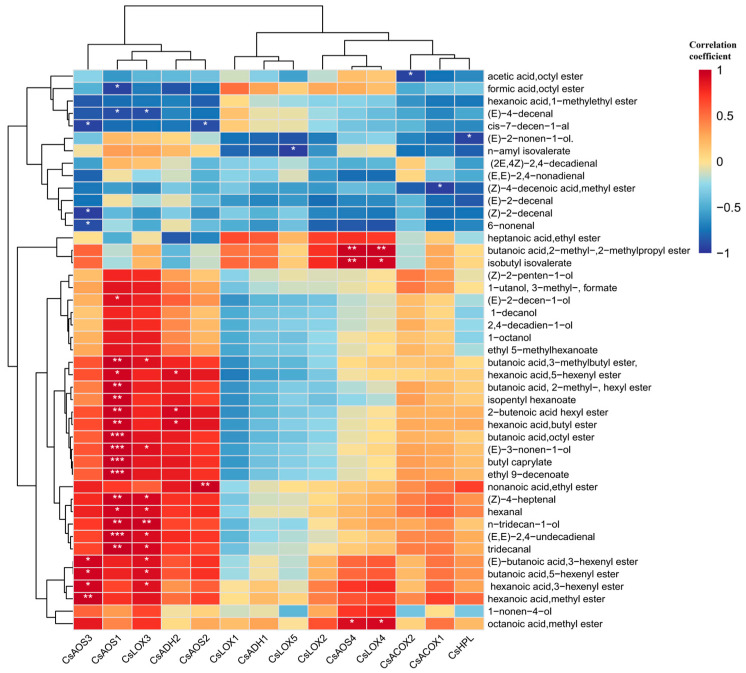
The correlation analysis between fatty acid related genes and VFADs. One or two asterisks denote a statistically significant difference (* *p* < 0.05; ** *p* < 0.01; *** *p* < 0.001) in relative amount.

## Data Availability

The original contributions presented in this study are included in this article/Appendix A. Further inquiries can be directed to the corresponding author.

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
