# Peer review of "Dynamic Change of Volatile Fatty Acid Derivatives (VFADs) and Their Related Genes Analysis during Innovative Black Tea Processing"

_foods, 2024, doi:10.3390/foods13193108_

Round 1

Reviewer 1 Report

Comments and Suggestions for Authors

Report on the manuscript foods-3167122 entitled: Dynamic change of volatile fatty acid derivatives (VFAD) and their related genes analysis during innovative black tea (IBT) processing.

-          IBT is a processing protocol established by previous studies. I would not use such a nomenclature in the title since the readers cannot be familiar with it.

-          In the title, it can be read “dynamic change of VGAD”. However, in the Conclusions section, only comments regarding gene, transcriptomics, gene expression, etc. are shown.

Besides, phrases such as “elucidating mechanisms”, “elucidating dynamics”, “elucidating relationships” are used in the manuscript. However, no “elucidation” has been carried out.

Please, reconsider and discuss.

-          L. 160-222. The description of the results must be improved.

The description of the results is poor and lacks scientific rigor. The authors should improve the description with actual numerical values such as % difference.

In addition, the results are sometimes described as time dependent effects and others as “treatments” related to processing. In fact, in the title it can be read “dynamic change”.

The changes are related to the processing steps. Therefore, could they be considered as “dynamic change”?

Please, reconsider and discuss.

-          Figures 3, 4 and 5: SD bars? Statistical analysis?

Please, improve the description of the Results!

-          L. 86. Delete the reference 33 since it has been retracted.

-          L. 97-103. Please, rewrite avoiding the use of “We”.

-          Please, use the nomenclature E/Z instead of cis/trans.

-          Figure 1. It should say “linoleic” not “linolenic” in the bottom left orange globe.

Comments on the Quality of English Language

Several grammar errors can be detected in the text.

For example, 

- L. 51. "difered"?

- L. 53-54. Rewrite. "These adjustment"? "Which wins"?

Author Response

COMMENT 1: IBT is a processing protocol established by previous studies. I would not use such a nomenclature in the title since the readers cannot be familiar with it.

Response 1: Thank you for your suggestion. We have made revisions. The manuscript no longer uses IBT. We have replaced “IBT” with “innovative black tea”.

COMMENT 2:  In the title, it can be read “dynamic change of VFAD”. However, in the Conclusions section, only comments regarding gene, transcriptomics, gene expression, etc. are shown.

Response 2: Thank you for your suggestion.We have supplemented and improved the relevant content on metabolite VFAD in the conclusion section. (Line 468-471)

COMMENT 3:  Besides, phrases such as “elucidating mechanisms”, “elucidating dynamics”, “elucidating relationships” are used in the manuscript. However, no “elucidation” has been carried out.

Please, reconsider and discuss.

Response 3: Thank you for your reminder. Our use of 'elucidation' in wording is not appropriate. We have changed 'elucidate' to 'illuminate', 'explain' or 'explain', which is more in line with our research level.

COMMENT 4: L. 160-222. The description of the results must be improved.

The description of the results is poor and lacks scientific rigor. The authors should improve the description with actual numerical values such as % difference.

Response 4: Thank you for your suggestion. We have made significant revisions and improvements in this section, hoping to meet your requirements. (Line 181-269, highlighted in red)

COMMENT 5:  In addition, the results are sometimes described as time dependent effects and others as “treatments” related to processing. In fact, in the title it can be read “dynamic change”.

The changes are related to the processing steps. Therefore, could they be considered as “dynamic change”?

Please, reconsider and discuss.

Response 5: Thank you for your suggestion. The primary manufacturing process of innovative black tea includes withering, turn-over, rolling, fermentation, and drying. The withering and fermentation are mainly carried out through long-term standing (indoor natural withering & solar-withering, static fermentation). After creating suitable temperature and humidity conditions, without excessive personification techniques, the transformation, formation, and accumulation of compounds are completed during the static period (withering for more than 8 hours, fermentation for about 3-5 hours) [1-2]. Therefore, we tend to have a certain degree of time-dependent effects in some descriptions. As for the expression 'dynamic change', my research group has also expressed it in previous research papers on tea processing and quality [3-5]. At the same time, other related studies (not in our research group) have also made similar statements [6-7].

Ref:

[1] Ntezimana, B.; Li, Y.; He, C.; Yu, X.; Zhou, J.; Chen, Y.; Yu, Z.; Ni, D. Different Withering Times Affect Sensory Qualities, Chemical Components, and Nutritional Characteristics of Black Tea. Foods 2021, 10, 2627

[2] Zhu, J.; Wang, J.; Yuan, H.; Ouyang, W.; Li, J.; Hua, J.; Jiang, Y. Effects of Fermentation Temperature and Time on the Color Attributes and Tea Pigments of Yunnan Congou Black Tea. Foods 2022, 11, 1845.

[3] Zi-wei Zhou, Qing-yang Wu, Zhi-ling Yao, Hui-li Deng, Bin-bin Liu, Chuan Yue, Ting-ting, Deng, Zhong-xiong Lai, Yun Sun*. Dynamics of ADH and related genes responsible for the transformation of C6-aldehydes to C6-alcohols during the postharvest process of oolong tea. Food science & nutrition, 2020,8(1): 104-113.

[4] Zhou Zi-Wei,Wu Qing-Yang,Yang Yun,Hu Qing-Cai,Wu Zong-Jie,Huang Hui-Qing, Lin Hong-zheng, Lai Zhong-xiong, Sun Yun*. The Dynamic Change in Fatty Acids during the Postharvest Process of Oolong Tea Production[J]. Molecules, 2022, 27(13): 4298-4298.

[5] Ziwei Zhou†, Qingyang Wu†, Hongting Rao, Liewei Cai, Shizhong Zheng, Yun Sun*. The Dynamic Change in Aromatic Compounds and Their Relationship with CsAAAT Genes during the Post-Harvest Process of Oolong Tea. Metabolites, 2023, 13: 868.

[6] Genome-wide epigenetic dynamics of tea leaves under mechanical wounding stress during oolong tea postharvest processing

[7] Dynamics of Water-soluble Pectin in the Roasting Process during Green Tea Manufacturing

COMMENT 6:  Figures 3, 4 and 5: SD bars? Statistical analysis? Please, improve the description of the Results!

Response 6: Thank you for your reminder. We have added SD bars and statistical analysis in Figures 3, 4 and 5. Based on statistical analysis, we have optimized and improved the description of the results section. (Line 181-269, highlighted in red)

COMMENT 7:  L. 86. Delete the reference 33 since it has been retracted.

Response 7: Thank you for your reminder. We sincerely apologize that we did not notice this issue at the time and we have deleted the reference 33.

COMMENT 8:  L. 97-103. Please, rewrite avoiding the use of “We”.

Response 8: Thank you for your suggestion. We have removed 'we' and revised the sentence structure. (Line 102-111)

COMMENT 9:  Please, use the nomenclature E/Z instead of cis/trans.

Response 9: Thank you for your reminder. We have replaced 'cis/trans' with 'E/Z' in the manuscript. (Line181、Line187)

COMMENT 10:  Figure 1. It should say “linoleic” not “linolenic” in the bottom left orange globe.

Response 10: Thank you for your detailed suggestions. We have made corrections to the incorrect vocabulary in Figure 1 and the manuscript. As you said, the correct precursor for fatty acids should be α-linolenic and linoleic. ( Line 97)

COMMENT 11:  Comments on the Quality of English Language

Several grammar errors can be detected in the text.

For example,

- L. 51. "difered"?

- L. 53-54. Rewrite. "These adjustment"? "Which wins

Response 11: Thank you for your reminder. We have made corrections to the grammar issues in sections Line 51 and Line 53-54. At the same time, we also commissioned native English speakers to conduct inspections, carefully checking and correcting grammar issues throughout the entire manuscript.

Reviewer 2 Report

Comments and Suggestions for Authors

In this study, a total of 45 VFAD, comprising 11 derived aldehydes, 9 derived alcohols, and 25 derived esters were identified and their dynamic variations were observed. Transcriptome analysis was performed to identify genes involved in the LOX pathway, resulting in the identification of CsLOX genes, 1 hydrogen peroxide lyase (CsHPL) gene, 11 alcohol dehydrogenases (CsADH) genes, 11 genes as acyl CoA oxidase (CsACOX) genes, and 3 allene oxide synthase (CsAOS) genes. Additionally, the expression levels of these genes were measured. In addition, qRT-PCR verification and correlation analysis were conducted to elucidate the relationship between VFAD and candidate genes. The manuscript is interesting and it needs minor revision according to the following remarks.

At the end of Introduction part there is need to point out what is the novelty of present study. What was done for the first time? Namely, it is written that „..aim was to explore the formation mechanism of aroma compounds in IBT, thereby providing valuable insights for targeted manufacturing and quality enhancement of black tea.“, but there is no indication of the novelty.

There is need to apply IUPAC rules for the name of the compounds, e.g. (E)-dec-2-enal instead of (E)-2-decenal, (E,E)-undeca-2,4-dienal instead of (E,E)-2,4-undecadienal, than it is better to change cis- and trans- to (E)- or (Z)- to be uniform in overall text

In the results there is need to add the table of identified compounds with the corresponding retention indices. In addition, there is need to describe the quantization process of the volatiles as well as the standrads used should be mentioned in the experimental part.

In the conclusion there is need to refer to the identified volatile organic compounds.

Comments on the Quality of English Language

The English grammar should be rechecked, some minor errors are present.

Author Response

COMMENT 1: At the end of Introduction part there is need to point out what is the novelty of present study. What was done for the first time? Namely, it is written that „..aim was to explore the formation mechanism of aroma compounds in IBT,  thereby providing valuable insights for targeted manufacturing and quality enhancement of black tea.“, but there is no indication of the novelty.                                       

Response 1: Thank you for your suggestion. We have added an indication of the novelty in the introduction section.  (Line 107-110)

COMMENT 2: There is need to apply IUPAC rules for the name of the compounds, e.g. (E)-dec-2-enal instead of (E)-2-decenal, (E,E)-undeca-2,4-dienal instead of (E,E)-2,4-undecadienal, than it is better to change cis- and trans- to (E)- or (Z)- to be uniform in overall text

Response 2: Thank you for your reminder. We have replaced 'cis/trans' with 'E/Z' in the manuscript. (Line181、Line187)

COMMENT 3: In the results there is need to add the table of identified compounds with the corresponding retention indices. In addition, there is need to describe the quantization process of the volatiles as well as the standrads used should be mentioned in the experimental part.

Response 3: Thank you for your suggestion.We have listed and explained the identified compounds with the corresponding retention indices (Table S3, Line 179, Line 484-485). We have added an explanation of the identification standard method in  the experimental part. (Line 147-151)

COMMENT 4: In the conclusion there is need to refer to the identified volatile organic compounds.

Response 4: Thank you for your suggestion. We have supplemented and improved the relevant content on metabolite VFAD in the conclusion section. (Line 468-471)